# Adenosine Receptors Profile in Fibromuscular Dysplasia

**DOI:** 10.3390/biomedicines10112831

**Published:** 2022-11-06

**Authors:** Claire Guiol, Sarah El Harake, Julien Fromonot, Mohamed Chefrour, Marguerite Gastaldi, Yassine Alibouch, Maxime Doublier, Pierre Deharo, Gabrielle Sarlon, Marion Marlinge, Nathalie Lalevee, Régis Guieu, François Silhol

**Affiliations:** 1Centre for Cardiovascular Research and Nutrition, C2VN, INSERM, INRAE, Aix Marseille University, 13000 Marseille, France; 2Hypertension Department, Pôle Cardio-Vasculaire, Timone University Hospital, 13000 Marseille, France; 3Laboratory of Biochemistry, Timone University Hospital, 13000 Marseille, France; 4Department of Cardiology, Timone University Hospital, 13000 Marseille, France

**Keywords:** adenosine, A_2B_ adenosine receptors, fibromuscular dysplasia

## Abstract

Fibromuscular dysplasia (FMD) is a non-inflammatory vascular disease that is characterized by unexplained systemic hypertension occurring in young people, associated with arterial stenosis, aneurysm rupture, intracranial/renal infarction, and stroke. Although the gold standard for the diagnosis remains catheter-angiography, biological markers would be helpful due to the delay from first symptom to diagnosis. Adenosine is an ATP derivative, that may be implicated in FMD pathophysiology. We hypothesized that changes in adenosine blood level (ABL) and production of adenosine receptors may be associated with FMD. Using peripheral blood mononuclear cells, we evaluated A_1_, A_2A_, and A_2B_ receptor production by Western blot, in 67 patients (17 men and 50 women, mean (range) age 55 (29–77) years and 40 controls, 10 men and 30 women, mean (range) age 56 (37–70)). ABL was evaluated by liquid chromatography, mass spectrometry. ABL was significantly higher in patients vs. controls, mean (range): 1.7 (0.7–3) µmol/L vs. controls 0.6 (0.4–0.8) µmol/L (+180%) *p* < 0.001. While A_1_R and A_2A_R production did not differ in patients and controls, we found an over-production of A_2B_R in patients: 1.70 (0.90–2.40; arbitrary units) vs. controls = 1.03 (0.70–1.40), mean + 65% (*p* < 0.001). A_2B_R production with a cut off of 1.3 arbitrary units, gives a good sensitivity and specificity for the diagnosis. Production measurement of A_2B_R on monocytes and ABL could help in the diagnosis, especially in atypical or with poor symptoms.

## 1. Introduction

Fibromuscular dysplasia (FMD) is an idiopathic systemic vascular disease, affecting mostly middle-aged women (mean age at diagnosis 46–53 years) and accounting for 10% to 20% of the cases of renal artery stenosis [1,2]. FMD is a segmental, non-atherosclerotic, non-inflammatory arterial disease, involving small and medium caliber arteries, mostly renal and carotid arteries but it can affect however other arterial territories. FMD is expressed in several ways: it can lead to arterial stenosis with a typical “string of beads” appearance in the most prevalent FMD multifocal form but is also associated with other arterial expressions: aneurysm, dissection, or tortuosity. This plurality of expression of the disease explains the different clinical expressions [3]. FMD classically presents as renovascular hypertension but can also manifest as stroke, aneurysm rupture, or unexpected myocardial infarct in young adults [2]. FMD is primarily a stenotic disease with lesions classified by computed tomography (CT), magnetic resonance imaging (MRI), or angiographic images. The diagnostic often occurs late in the life of patients due to the delay between the first symptoms and the diagnosis [4]. While the gold standard for the diagnosis remains medical imaging, the diagnosis may be difficult in the absence of characteristic lesions or multifocal involvement. Early diagnosis and treatment are important for a long-term prognosis and for the family screening. In this context, the use of new biomarkers will be helpful. The etiology of FMD remains unknown in spite of extensive research. A genetic ground has been suspected due to the presence of FMD in twins [5] while the prevalence of familiar case remains around 10% [6]. While no inflammatory process was found to be associated with FMD, a susceptibility gene (PTGIR, a gene encoding a prostaglandine I2 receptor) was suspected to be associated with FMD [7].

High level of nuclei progesterone receptors in dysplasic renal arteries of FMD was reported, suggesting that progesterone may play a role in the disease and could partly explain the sex ratio [8]. Finally, a plasma proteogenomic signature was recently advocated [9], which concerns a great number of molecules making it difficult to use these markers on a daily basis in everyday practice.

Recently a possible implication of the adenosinergic system has been advocated [10].

Adenosine is a ubiquitous nucleoside that mainly comes from the dephosphorylation of ATP. While a part of adenosine metabolism is associated with the methionine cycle.

Adenosine release occurs more particularly during ischemia, hypoxia, or inflammation [11,12].

Adenosine strongly impacts the vascular tone via its 4 G-coupled membrane receptors named A_1_R, A_2A_R, A_2B_R, and A_3_R [11]. While the activation of A_1_R leads mostly to vasoconstriction [13], the vasodilating effects are mainly secondary to the activation of A_2A_R [14,15] and A_2B_R [16]. The vasodilating effect occurs via K_V_, K_ATP_ channels [17]. Conversely, the activation of A_1_R can induce vasoconstriction via a direct (cAMP independent) effect and the activation a protein kinase C (PKC) [13]. It was also shown that adenosine leads to vasodilation by acting via endothelium-dependent NO release [18]. The relaxation of smooth muscle cells decreases vascular resistance favoring dioxygen delivery.

In FMD the vascular flow is disturbed in the affected areas notably with a decrease in renal perfusion, accounting for hypertension. Thus, the decrease in blood flow with a relative ischemia may activate the adenosinergic system to improve blood flow as it was described in coronary artery disease [19].

We hypothesized that the adenosinergic system may be disturbed in this affection.

The aim of this study was to evaluate adenosine blood level, and adenosine receptors production by FMD patient peripheral blood mononuclear cells (PBMC), these cells being a good model for the evaluation of the adenosinergic system in cardiovascular diseases [20,21,22].

## 2. Materials and Methods

### 2.1. Patients

Adult patients with multifocal FMD (diagnosed using CT scan or duplex ultrasound) were included according to current criteria [2,3]. Control group, matched for age and sex, was recruited among the medical staff.

### 2.2. Inclusion Criteria for Patients

We included (between October 2020 and November 2021), 67 patients diagnosed with multifocal fibromuscular dysplasia (FMD group) in our center of competence for rare vascular diseases (Timone University Hospital, Marseille, France) and 40 controls. The patients included were all referred to our center of vascular medicine and arterial hypertension. Criteria were as follows:Patients aged between 18 to 80 years.Patients with multifocal FMD of a renal or cervical artery with typical aspect “ string of beads “ appearance.Patients willing to participate in the study.

### 2.3. Inclusion Criteria for the Control Group

Subjects aged between 18 to 80 years.Subjects without history of personal or familial fibromuscular dysplasia.Subjects without story of cardiovascular disease or inflammatory process.Subjects free from medical treatment.Subjects willing to participate in the study.

### 2.4. Exclusion Criteria

Patients less than 18 or more than 80 years of age.Renal failure with GFR < 30 mL/min.Pregnant women.Patients with symptomatic atheromatous cardiovascular disease.Patient with inflammatory or chronic disease.Patient with a history of cancer or active cancer.Patient refusing to participate in the study.

### 2.5. Adenosine Blood Level (ABL) Measurement

#### 2.5.1. Blood Samples Collection

Whole blood was collected using finger puncture followed by deposit of a drop of blood (20 μL) on a blotting paper (Whatman 903 protein saver cards™) and dried over night at room temperature to obtain dried blood spot. Then, the blot spots were stored at room ambiance in a dried room, until analysis. Adenosine dosages were carried out in series of 10 patients and 5 controls.

#### 2.5.2. Blood Samples Extraction

The method has been previously described [23]. Briefly, six mm of blood spot were cut out followed by extraction (mix of methanol and internal standard) for 90 min at 45 °C. After extraction, aliquots (350 µL) were transferred into a new 2 mL safe-lock tube and evaporated to dryness at 60 °C under nitrogen; 150 μL of 0.1% formic acid in water were added and quickly vortexed before transferring into an HPLC auto sampler vial.

#### 2.5.3. Adenosine Concentration

After extraction, adenosine concentration was measured by LC-MS/MS as previously described [23]. Samples were analyzed using a Shimadzu UFLC XR system (Shimadzu, Marne la Vallee, France). The LC system was interfaced with an ABSciex 4500 triple quadrupole mass spectrometer (Shimatzu, Les Ulis, France) operating with an electrospray ionization source (ESI) using nitrogen (purity: 99.99%). About 10 µL of the extracted sample were injected onto a 2.1 × 100 mm, 3 μm column (Waters, Guyancourt, France). The mobile phase consisted of 3% methanol and 97% acidified water (0.1% formic acid) with a flow of 0.7 mL/min for 3.5 min. The gradient of methanol was increased to 30% for 3 min. The column was re-equilibrated for 2 min to starting conditions. In these conditions, the intra or inter-assay coefficient of variation was less than 10% (5 to 8%).

### 2.6. Expression of Adenosine Receptors in PBMC

PBMCs were chosen because pharmacological profile of their A_2A_R expressed at the surface, mirrors that of A_2A_R associated with cardiovascular tissues [20,21,22]. PBMCs were isolated from whole blood using the Vacutainer-CPT system (Becton-Dickinson, Franklin Lakes, NJ, USA, 8 mL) according to the manufacturer’s instructions. Briefly, blood samples were centrifuged (1500× *g*, 30 min) within 2 h of collection. After centrifugation, mononuclear cells were collected from the plasma/Ficoll interface and washed twice with phosphate buffered solution. The cell pellet was stored at −80 °C until used for Western blot analysis.

#### Western Blots

The procedure for Western blotting has been described [24,25,26,27,28]. Briefly, A_1_R, A_2A_R, and A_2B_R expressions of PBMCs were determined by Western blot using Adonis, an agonist-like monoclonal antibody to human A_2A_R [28], a rabbit polyclonal antibody to human A_1_R (ab82477, Abcam^®^), and a goat polyclonal antibody to human A_2B_R (ab40002, Abcam^®^). PBMC pellets (0.25 × 10^6^ cells) were solubilized using lysis buffer and sonication. Samples were then submitted to standard 12% polyacrylamide gel electrophoresis under reducing conditions before transferring to a polyvinylidene difluoride membrane. The filters were then incubated with Adonis (1 µg/mL) or A_1_R or A_2B_R antibodies (0.25 µg/mL). Blots were revealed by phosphatase alkaline-labeled antispecies as second antibodies and adapted colorimetric substrate. Each band corresponding to the adenosine receptors (36 kDa for A_1_R, 45 kDa for A_2A_R, and 37 kDa for A_2B_R) was submitted to densitometry analysis using the Image J 1.42q software (National Institutes of Health). Results were expressed in arbitrary units (AU), defined as the ratio of pixels generated by the adenosine receptor band to pixels generated by the background signal, as previously described [24,25,26,27]. The assays were carried out in series of 10 patients and 5 controls. Blot papers were stored at room temperature no more than one month. For Western blot, the cell pellets were stored at −80 °C until assay. All blotting were performed in duplicate. In these conditions, the mean intra or inter-assay coefficient of variation was less than 10%.

## 3. Statistical Analysis

Statistical analyses were performed using GraphPad Prism program version 8.4.3. All statistical tests were two-sided and *p*-values < 0.05 were considered statistically significant. Quantitative and qualitative variables were compared using Mann–Whitney U tests, Chi-square test, or Fisher’s exact test, as appropriate. Qualitative variables were reported as median with minimal and maximal range (min-max range). Diagnostic test accuracy of adenosine blood level and A_2A_B receptor expression in PBMCs for fibromuscular dysplasia was assessed by receiver operating characteristic (ROC) curves and area under the receiver operating characteristic (AUROC). AUROC from 0.9 to 1 was considered as excellent accuracy, 0.8 to 0.9 as good, 0.7 to 0.8 as good, 0.6 to 0.7 as good, and <0.6 as insufficient. The Youden index was assessed for all point of ROC curves (sensitivity + specificity − 1) and the optimal threshold was determined by the maximum value of the Youden index. We hypothesized that a variation of at least 50% in ABL and receptor expression level will have physiological consequences. In this perspective, a number of 50–60 patients vs. 40 controls appears sufficient.

## 4. Results

### 4.1. Patients

A total of 67 patients, mean age and range 55 (29–77) years, and 40 controls 56 (37–70) years were included. Clinical characteristics of patients are given in Table 1. Patients were 17 men (25.4%) and 50 women (74.6%) and controls were 10 men (25%) and 30 women (75%). No significant differences occurred between patients and controls concerning age (*p* = 0.73) and sex ratio (*p* = 0.97).

### 4.2. Adenosine Blood Level (ABL)

ABL was significantly higher in patients vs. controls (see Figure 1).

Finally, while smoking may alter nucleosides metabolism in vitro (29), no difference appeared when considering current smokers (*n* = 25) vs. no smokers (*n* = 42) 1.85 µM vs. 1.7 µM, *p* = 0.8.

### 4.3. Adenosine Receptors Production in PBMCs

While no significant difference occurred concerning A_1_R or A_2A_R production in patients vs. controls: A_1_R (A.U.): Patients = 0.97 (0.78–1.20) vs. Controls = 0.90 (0.80–1.15) (*p* = 0.17). A_2A_R: Patients = 1.30 (0.90–1.80) vs. Controls = 1.29 (0.90–1.50) (*p* = 0.06);

Patients exhibit a higher A_2B_R (A.U) production level compared with controls: 1.70 (0.90–2.40) vs. Controls = 1.03 (0.70–1.40), mean + 65% (*p* < 0.001) see Figure 2.

While cigarettes smoking may modify nucleotides metabolism [29], no difference appeared when considering current smokers vs. no smokers:

A_1_R: 0.9 AU vs. 0.98 *p* = 0.5; A_2A_R: 1.3AU vs. 1.3AU *p* = 0.49; A_2B_: 1.7 AU vs. 1.85AU, *p* = 0.5. There was no significant difference between women and men patients in ABL: median 1.7 vs. 1.9 *p* = 0.3 and in receptors expression: A_1_R: 0.9 vs. 1, *p* = 0.08; A_2A_R = 1.3 vs. 1.4 *p* = 0.052; A_2B_R: 1.7 vs. 1.8 *p* = 0.07. No difference appears between women and men in controls (data not shown). Finally, a correlation was found between ABL and A_2B_ receptor expression (Figure 3).

Receiver-operating characteristic (ROC) curves show a good sensitivity/specificity ratio for ABL > 0.85 µmol/L (see Figure 4A). For A_2B_R production, ROC curves show a good sensitivity/specificity ratio with a cut off >1.23 AU (see Figure 4B). Finally, when combining ABL and A_2B_R expression, ABL > 0.85 µmol/L and A_2B_R production > 1.23 AU give a sensitivity of 89.6% and a specificity of 100%.

## 5. Discussion

The main result of this study is that while no significant modifications in A_1_R or A_2A_R expression was found in patients with FMD, we report here an overexpression of A_2B_ receptors in PBMC in a context of high ABL. Abnormalities in ABL and adenosine receptors expression have been reported in some cardiovascular diseases. Thus, high ABL have been found in chronic cardiac failure (CCF, [30,31]), in vasovagal syncope (VVS, [32,33]), while A_2A_R overproduction has been reported in VVS patients [32,33] or during cardiogenic shock [34]. Interestingly, while high A_2A_R and high ABL in VVS seem to participate in the cause of the disease, since high A_2A_ production and a specific polymorphism in A_2A_R gene are associated with VVS suggesting a genetic predisposition [35], the increase in these parameters in CCF or in cardiogenic shock seems rather an adaptive mechanism. Both high ABL and A_2B_R overproduction have been reported in patients with atrial fibrillation (AF) [36]. However, if the increase in A_2B_ production participates in the AF process or if conversely it is an adaptive mechanism that remains unknown. Whatever, in AF patients, the overproduction of A_2B_R, while significantly remained weak, around 15%. In this study, we described an overproduction of 65% in mean associated with a high ABL level, +180% compared with controls. ROC curves indicate a good predictive value for the A_2B_R production with a cut-off 1.23 associated with a good sensitivity of A_2B_R overproduction. Similar data have previously been reported in a pilot study including a weak number of FMD patients that did not permit to establish ROC curves [10]. 

Both A_2A_ and A_2B_ receptors are implicated in the control of blood flow via the regulation of vascular tone and muscle cell proliferation [37,38]. Vascular smooth muscle cell proliferation is an important component of vascular remodeling. Adenosine is a candidate as a regulator of smooth muscle cell proliferation via A_2B_R [39,40]. Indeed, pharmacological approaches demonstrate that activation of the A_2B_R leads to inhibition of vascular smooth muscle cell proliferation [39,40]. In vitro studies demonstrated that A_2B_R inhibits smooth muscle cell proliferation mediated by B-Myb regulation of the A_2B_R [41].

Activation of A_2B_R protects against experimental vascular injury. Thus, A_2B_-deficient mice (KO) enhances post injury neo-intima formation in the vasculature in an experimental model of femoral injury [42]. Interestingly high level of TNF alpha, an up regulator of CXCR4 and of vascular smooth muscle cells (VSMC) proliferation was reported in this model of KO mice [41]. Among adenosine receptors, the A_2B_R have the lower affinity for adenosine and thus can be activated only in the case of high adenosine extra cellular level, like in the present study.

FMD is associated with ischemia in a lot of territories including renal arteries [1], coronary arteries, vertebral arteries, carotids [2,3]. The cause of ischemia is different from other blood vessels disorders that affect arteries such as atherosclerosis or thrombosis. The ischemia occurs due to the presence of different artery abnormalities such as narrowing, beaded appearance or during complications such as aneurysms, or arteries dissection.

Ischemia promotes adenosine release, explaining the very high ABL (until 3 µM in some patients). This increase is sufficient to activate A_2B_ low affinity receptors. A_2_ adenosine receptor subtypes have been described as sensors of tissue damage and their activation leads to anti-inflammatory properties more particularly in vessels. Activation of A_2A_R and A_2B_R leads to the inhibition of pro-inflammatory cytokine release [43] and inhibits macrophage migration [44]. Activation of A_2B_R may have antifibrotic effects [45], however pro-fibrotic action have also been reported and the role of A_2B_R remains controversial [45]. The increase in ABL however does not always have beneficial effects. Indeed, deleterious effects of prolonged exposure to high extracellular adenosine levels have been suspected in cancer, auto-immunity, neurodegenerative diseases, diabetes, and inflammatory conditions [46]. In the cardiovascular system, while acute high adenosine blood level may have cardioprotective effects by protecting heart and vessels, chronic high ABL may have deleterious effects [12]. Indeed, it was reported that activation of A_2B_R by high extracellular adenosine level may promote myocytes apoptosis in the vascular walls [47].

The well-known presence of a sex ratio in FMD [3] indicates that the high ABL and high A_2B_ receptor expression is probably a consequence of the disease rather than a cause. Indeed, there is no evidence for a difference in expression of adenosine receptors pending on sex. In this study, no significant difference in ABL or in receptors expression was found between women and men in both patients and controls. The ischemia in arterial territories leads to the release in adenosine that in turn up-regulates A_2B_ receptors which have low affinity [48]. This could be an attempt at repermeabilization to dilate the vessels via activation of A_2_ receptors. Note that while not significant there is a trend in an increase in A_2A_R production in FMD patients (*p* = 0.06).

Outside the cardiovascular system, an up-regulation of A_2B_ receptors has been reported on monocytes in response to parasite [49] and in cancer [50]. Gamma interferon is also susceptible to induce an up-regulation of A_2B_R, a mechanism by which macrophages can be deactivated [51].

## 6. Study Limitation

Although involving a limited number of patients, our study seems to indicate that the expression level of A_2B_ receptors could help in the diagnosis of FMD. A larger number of patients may be needed to establish ROC curves with sufficient sensitivity and specificity.

We only evaluated the level of expression but not the function (i.e., production of cAMP) by mononuclear cells. We therefore cannot know whether produced in abundance these receptors are functional. Here too lies the limit of our study. However, our study originally falls more within the development of new markers. Our study does not allow us to conclude whether A_2B_ receptor modulating agents could be used in the treatment of the condition. 

We cannot exclude the presence of asymptomatic FMD in the control group. However, the prevalence of FMD in France is less than 1/259 [52]. Thus, the probability to include asymptomatic patients in the control group is very weak. Finally, it should be better to include more controls. However, in our study in spite of the weak number of controls, the difference in ABL and in adenosine receptors expression is stark and significant.

Finally, we cannot exclude, in FMD patients, abnormal activities of enzymes that metabolize adenosine. 5′-nucleotidases activities may be higher or adenosine deaminase activity lower. Perhaps rate of erythrocyte ATP degradation into AMP is higher in patients. The answer to these questions needs further investigations.

## 7. Conclusions

We found that A_2B_ overproduction and high ABL are associated with FMD outside an inflammatory context. Thus, expression measurement of A_2B_R and ABL could help in the diagnosis, in particular, in especially atypical forms or those with very few clinical signs. This is all the more so since, apart from cancers, a such hyperexpression of A_2B_ receptors has been rarely mentioned. However, if the overproduction of A_2B_ in FMD participate into the disease progression or if it is an adaptive mechanism needs further investigations.

## Figures and Tables

**Figure 1 biomedicines-10-02831-f001:**
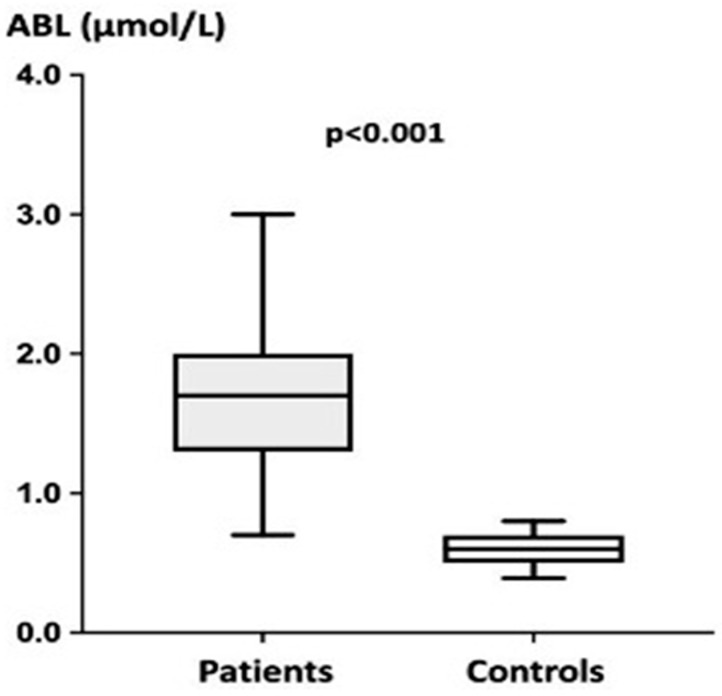
Comparison of adenosine blood levels in patients with fibromuscular dysplasia (*n* = 67) and healthy subjects (*n* = 40). Patients: (**median and range**)**:** 1.70 (0.7–3.0) µM vs. Controls = 0.60 (0.4–0.8) µM, *p* < 0.001.

**Figure 2 biomedicines-10-02831-f002:**
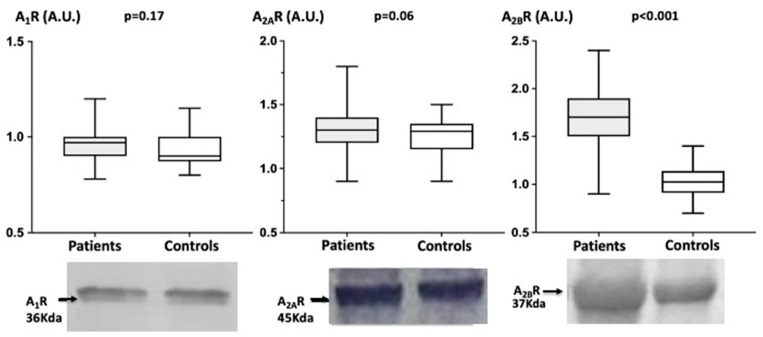
Comparison (median and range) of adenosine receptors production in PBMCs in patients with fibromuscular dysplasia (*n* = 67) and healthy subjects (*n* = 40). Original blots are given in supplement file.

**Figure 3 biomedicines-10-02831-f003:**
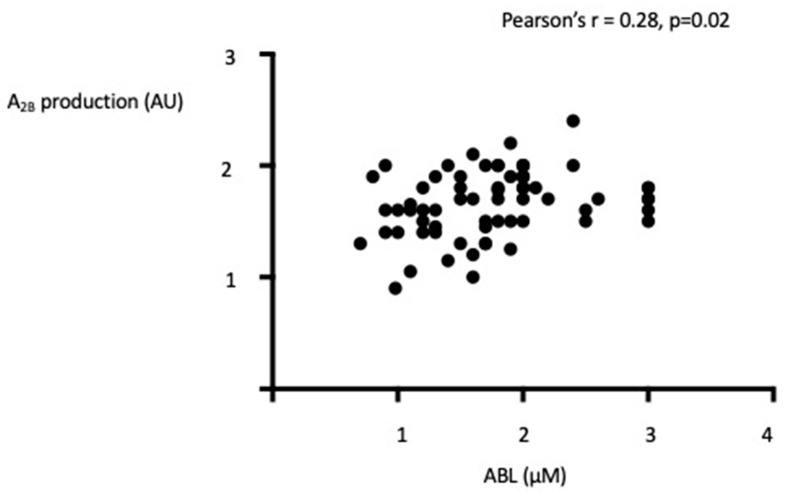
Correlation between A_2B_ production and ABL in 67 patients with fibromuscular dysplasia.

**Figure 4 biomedicines-10-02831-f004:**
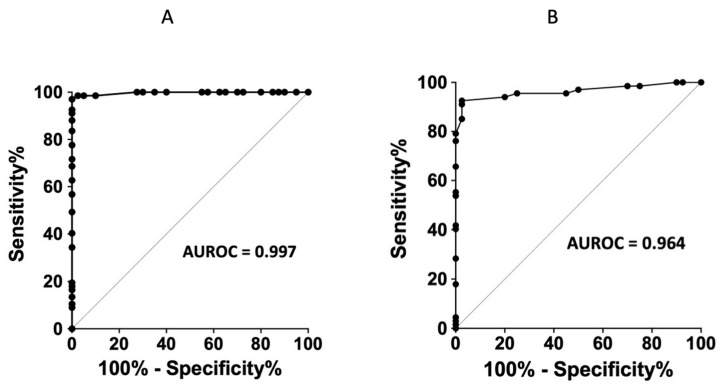
(**A**). Receiver-operating characteristic (ROC) curve of adenosine blood level. ROC—ABL: AUROC (CI_95%_) = 0.99 (0.99–1.00); *p* < 0.001. Threshold determination: ABL > 0.85 μmol/L: Se 97.0% (89.8–99.5%). Sp 100.0% (91.2–100.0%). (**B**). Receiver-operating characteristic (ROC) curve of A_2B_R expression in PBMCs.

**Table 1 biomedicines-10-02831-t001:** Main clinical characteristics of patients. ARBs: angiotensin receptor blockers. ACE inhibitors: Angiotensin converting-enzyme-inhibitors.

	FMD *n* = 67	Controls *n* = 40
**Age (Years, mean, Range)**	55 (29–77)	56 (37–70)
M/F	50/17	30/10
**Race**		
White	67	35
Black	0	1
Asian	0	4
Hypertension	29 (43%)	0
Current smoker	25 (37%)	12 (30%)
Hyperlipemia	32 (47%)	0
**Medications**		
Diuretic	21 (31%)	0
ARBs	20 (29%)	0
Beta-blockers	8 (12%)	0
ACE Inhibitors	15 (22%)	0

## Data Availability

Data are available on request. The raw data supporting the conclusions of this manuscript can be made available by the authors, without undue reservation.

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
