# Peer review of "Adenosine Receptors Profile in Fibromuscular Dysplasia"

_biomedicines, 2022, doi:10.3390/biomedicines10112831_

Round 1
Reviewer 1 Report
As the main result of this study, overexpression of A2B receptors found in patients with FMD, while no significant modifications in A1R or A2AR expression in a context of high ABL. Sihol et. al (doi.org/10.1038/s41440-019-0379-3) reported the same that FMD is associated with high A2BR expression. The current study is just an incremental advancement compared to previous study by Sihol et .al. and not suited for publication in Biomedicines.
Author Response
Dear Reviewer
Our team's previous work in 2020 was a pilot study as stated in lines 241-242, involving a small number of patients. It was important to confirm our results on a larger number of patients and especially to be able to establish a Cut off for A2B receptor expression and to be able to plot ROC curves indicating sensitivity and specificity. This is done with this work.
We hope that this explanation will convince you
R Guieu
Reviewer 2 Report
Fibromuscular dysplasia is a non-inflammatory vascular disease that is characterized by unexplained systemic hypertension occurring in young people. Fibromuscular dysplasia is primarily a stenotic disease with lesions classified by computed tomography, magnetic resonance imaging, and angiographic images. However, the diagnosis may be difficult in the absence of characteristic lesions or multifocal involvement. In this context, the use of new biomarkers will be helpful. Thus, the aim of this study was to evaluate adenosine blood level and adenosine receptors in peripheral blood mononuclear cells of fibromuscular dysplasia patient. The authors demonstrated that adenosine blood level was significantly higher in patients than control group. Among adenosine receptors, protein level of A2B receptor increased in peripheral blood mononuclear cells from patient. The authors conclude that expression measurements of A2B receptor and adenosine blood level may be diagnostic of fibromuscular dysplasia. The manuscript is well-written and the methods sound. However, English grammar and syntax in the manuscripts should be checked and corrected by a native English-speaking person.
Author Response
Thank you for your comments
The text was reviewed by a native English speaking, Tania Sharma
Reviewer 3 Report
In the paper of Guiol C. et al. authors studied adenosine level in peripheral blood and expression of adenosine receptors in monocytes of patients with fibromuscular dysplasia (FMD). They found that in patients with FMD adenosine level in blood is significantly increased and quantity of adenosine A2B-R is elevated comparing with that of controls. The authors suggest that this two changes can be good diagnostic markers of FMD.
Generally, the paper is well written and results are good presented.
I have only minor points to improve the paper:
1. Authors introduced abbreviation FMD for fibromuscular dysplasia, but then used also FDM for the same (pages 2, 9 and Table 1). Please be consistent.
2. Change name of section 2.5.3. Adenosine dosage to Adenosine concentration
3. Change name of section 2.6. PBMC Adenosine receptors production to Expression of Adenosine Receptors in PBMC
4. In line 171 Change 0.8 to 0.9 as good, 0.7 to 0.8 as good, to 0.7 to 0.9 as good,
5. In Table 1 move legend for ARBs from the title to under table
6. In section 4.2. and Figure 1 do not repeat the figures of ABL levels twice
7. In the legends of Fig.1-2 indicate what way figures are presented – Mean, SEM, etc.
Author Response
In the paper of Guiol C. et al. authors studied adenosine level in peripheral blood and expression of adenosine receptors in monocytes of patients with fibromuscular dysplasia (FMD). They found that in patients with FMD adenosine level in blood is significantly increased and quantity of adenosine A2B-R is elevated comparing with that of controls. The authors suggest that this two changes can be good diagnostic markers of FMD.
Generally, the paper is well written and results are good presented.
Answer : Thank You
I have only minor points to improve the paper:
- Authors introduced abbreviation FMD for fibromuscular dysplasia, but then used also FDM for the same (pages 2, 9 and Table 1). Please be consistent.
Answer: sorry you are wright. This has been corrected throughout the text
- Change name of section 2.5.3. Adenosine dosage to Adenosine concentration
Answer: this was done
- Change name of section 2.6. PBMC Adenosine receptors production to Expression of Adenosine Receptors in PBMC
Answer: this was done
- In line 171 Change8 to 0.9 as good, 0.7 to 0.8 as good, to 0.7 to 0.9 as good,
Answer: this was done
- In Table 1 move legend for ARBs from the title to under table
Answer: this was done
- In section 4.2. and Figure 1 do not repeat the figures of ABL levels twice
Answer: you are wright. This was corrected
- In the legends of Fig.1-2 indicate what way figures are presented – Mean, SEM, etc.
Answer: median and range are now specified
Hoping this version meets your approval
R Guieu; MD, PhD
Round 2
Reviewer 1 Report
I am convinced with the explanation by the authors.